# Actor-Critic Algorithm for High-dimensional PDEs

## Abstract

We develop a deep neural network model to solve high-dimensional nonlinear parabolic partial differential equations (PDE). Our model extends and advances the DBSDE model in the following 3 aspects: 1) the trainable parameters are reduced by $N$ times, where $N$ is the number of steps to discretize the PDE in time, 2) the model convergence rate is an order of magnitude faster, 3) our model has fewer tuning hyperparameters. . Our model is designed to maximally exploit the Markovian property of a BSDE system and utilizes an Actor-Critic network architecture. Our algorithm design leads to a significant speedup with higher accuracy level. We demonstrate the performance improvements with numerical experiments solving several well-known PDEs with dimensions on the order of 100.

## 1   Introduction

High Dimensional partial differential equations (PDEs) are encountered in many branches of modern sciences such as the Schrödinger equation for quantum many-body problem, the nonlinear Black-Scholes equation for pricing financial derivatives, and the Hamilton-Jacobi-Bellman equation for multi-agent game theories. In this work, we introduce a new model that effectively address those issues by exploiting the Markovian property of the BSDE system, which is rarely discussed in the literature. The Markovian property enables us to utilize an Actor-Critic neural network architecture in solving high dimensional PDE problems for the first time. Taking advantage of the variance reduction affect of Actor-Critic, our model is shown to make some significant performance improvements compared to existing deep learning based PDE solvers:

1. *largely reduced trainable parameters from $\mathcal{O}(Nd^2)$ to $\mathcal{O}(d^2)$:* here $N$ is the number of time steps that discretizes the temporal dimension, and $d$ is the spatial dimension of the PDEs. Namely, our algorithm is relieved from the constraint that the network complexity needs to scale linearly with the time steps and requires only a light-weight network. Therefore, calculating the gradients for all parameters is faster and consumes less memory.

2. *faster convergence rate:* In all the numerical experiments we studied, the convergence rate of our model is at least one order magnitude faster than DBSDE while giving the same (if not higher) level of solution accuracy. The fact that our algorithm requires less parameters and faster convergence rate leads to a significant run-time speed-up during training. For example, Quadratic Gradients equation is solved 18 times faster than that solved by DBSDE, and the Allen Cahn equation is 27 times faster than DBSDE.

3. *less hyperparameters to tune:* The existing deep learning based solvers need to prescribe an 1-d interval from which the initial solution is sampled. The range of the interval is defined by two hyperparameters. Numerically, we find that the convergence rate and solution accuracy are both sensitive to the choice of the two hyperparameters. Therefore, parameter tuning is a

necessity. By design, our model does not require such hyperparameters, which is partially attributed to the variance reduction affect of the Actor-Critic algorithm.

## 2   Approach

We start with the definition of a nonlinear parabolic PDE in the general form. Let $u\colon [0,\infty) \times R^d \to R$, $u \mapsto u(t,x)$ be the unknown vectorial function with the dimension $d$, we seek to find the value of $u$ at any given point $\xi$ such that it satisfies the following general nonlinear parabolic PDE:

$$\frac{\partial u}{\partial t} + \frac{1}{2}\mathrm{Tr}\left(\sigma(t,x)\sigma(t,x)^T(\mathrm{Hess_x u})\right) + \nabla u \cdot \mu(t,x) + f(t,x,u(t,x),\sigma^T(t,x)\nabla u(t,x)) = 0$$

(1)

with the terminal condition $u(T,x) = g(x)$. Here $t \in [0,T]$ and $x \in \mathbb{R}^d$ are the time and space variable respectively. $\mu(t,x) \in \mathbb{R}^d$ and $\sigma(t,x) \in \mathbb{R}^{d\times d}$ are known vector-valued functions. $\sigma^T$ is the transpose of $\sigma$. $\nabla u$ and $\mathrm{Hess}_x u$ represents the gradient and the Hessian of function $u$ w.r.t $x$. Tr denotes the trace of a $d \times d$ matrix. $f$ is a known scalar-valued nonlinear function. The goal is to find the solution $u(0,\xi)$ for some point $\xi \in \mathbb{R}^d$ at $t = 0$.

Following the same Feyman-Kac approach as in DBSDE model, we arrive at the equivalent discretized stochastic differential equations:

$$X_{t_{n+1}} = X_{t_n} + \mu(t_n, X_{t_n})(t_{n+1} - t_n) + \sigma(t_n, X_{t_n})\Delta W_{t_n}$$
$$u(t_{n+1}, X_{t_{n+1}}) = u(t_n, X_{t_n}) + Z_{t_n}^T \Delta W_{t_n} - f\left(t_n, X_{t_n}, u(t_n, X_{t_n}), Z_{t_n}\right)$$

(2)

where $Z_{t_n} = \left[\nabla u(t_n, X_{t_n})\right]^T \sigma(t_n, X_{t_n})$, and $\Delta W_{t_n} = W_{t_{n+1}} - W_{t_n}$. From the numerical point of view, (2) defines a controlled stochastic dynamics that can be efficiently sampled by simulating Brownian processes $W_{t_n}$, with $\mu$ and $\sigma$ given. Note that $N$ is a hyperparameter which needs to be tuned for different equations. The sensitivity study of $N$ is yet available in the literature.

A key feature that differentiates our model from others is that we exploit the Markovian property of the System (2). Therefore, we need to only deploy *one* multilayer feedforward network with batch-normalization, say $\theta_a$, to approximate $Z_{t_n}$ instead of a sequence of $N$ multilayer feedforward networks which is currently adopted by other deep learning based models. In addition, we parametrize $u(X_{t=0})$ with a second multilayer feedforward network, say $\theta_v$, while the other solvers use only one trainable parameter to represent the solution $u(X_{t=0})$ and train it together with the policy network. To some extent, $\theta_a$ is comparable to the policy network and $\theta_v$ to the critic network in model based reinforcement learning. A combination of such two networks within one framework is commonly referred to as the Actor-Critic algorithm.

To close the loop, we still need to define a loss function for training:

$$l(T) = \mathbb{E}\left[\left|g(X_T) - u(\{X_{t_n}\}_{0 \le n \le N}, \{W_{t_n}\}_{0 \le n \le N})\right|^2\right]$$

namely the loss function measures how close the predicted solution $u(T,x)$ matches the terminal boundary condition. In practice, to prevent the loss from blowing up, we clip the quadratic function by linearly extrapolating the function beyond a predefined domain $[-D_c, D_c]$, analogous to the trick used by Proximal Policy Optimization Schulman et al. (2017) which enforces a not-too-far policy update [1]. We use $D_c = 50$ in all of our experiments.

Given the temporal discretization above, the path $\{X_{t_n}\}_{0 \le n \le N}$ can be easily sampled using (2), the dynamics of which are problem dependent due to the $\mu$ and $\sigma$ terms. Fig. 1 illustrates a forward pass and a backward pass in one iteration where $\theta_v$ and $\theta_a$ are updated.

In terms of training process, $X_t$ and $W_t$ are sampled first by running the dynamics. We use $\theta_v$ to generate a guess, $u(X_{t=0})$ which is then passed forward in time to get $u(X_{t=T})$. The loss is backpropagated to update $\theta_v$ and $\theta_a$ with either stochastic gradient descent or other optimization methods alike. The total number of training steps is preset but we also find that using an early-stop mechanism usually leads to shorter run time while producing the same level of accuracy. [2]

---

[1]The difference is that PPO puts the constraint on the KL-divergence between consecutive updates instead of the least square measure.

[2]We do not use early-stopping in the numerical experiments as we want to have a fair comparison with other models in terms of run-time and convergence rate.

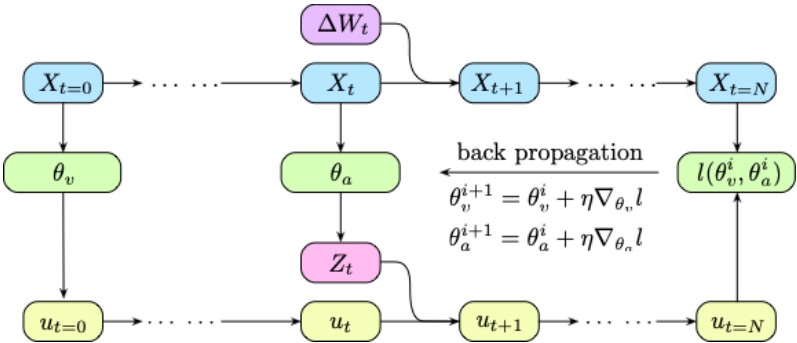

Figure 1: Forward and backward propagation of the $i_{\text{th}}$ iteration

Table 1: Run-time and Relative-error for all numerical examples

| PDE Examples | Run Time Actor-Critic | Run Time (DBSDE) | Relative Error Actor-Critic | Relative Error (DBSDE) |
|---|---|---|---|---|
| Hamilton Jacobi Bellman | 3 s | 22 s | 0.22% | 0.53% |
| Burgers Type | 20 s | 122 s | 3.4% | 0.31% |
| Reaction Diffusion | 132 s | 801 s | 0.61% | 0.69% |
| Quadratic Gradients | 9 s | 166 s | 0.06% | 0.08% |
| Allen Cahn | 5 s | 138 s | 0.25% | 0.46% |
| Pricing Option | 7 s | 20 s | 0.37% | 0.56% |

## 3 Preliminary Results

We solve the same set of examples presented by Weinan et al. (2017); Han et al. (2018). We also intentionally use the same computing environment settings in order to pinpoint the algorithm advantage. The run-time, relative error of the experiments we solved are presented in Table. 1 along side those in DBSDE.

### 3.1 Reduced trainable parameters

The number of trainable parameters in our algorithm, $\rho_0$, can be calculated as:

$$\rho_0 = 2 \times \underbrace{\left((d+10) + (d+10)^2 + d(d+10)\right)}_{\text{fully connected layers of } \theta_a \text{ and } \theta_v} + \underbrace{2(d+10) + 2(d+10) + 2d}_{\text{batch normalization layers of } \theta_a \text{ and } \theta_v} \tag{3}$$

In comparison, the number of trainable parameters of DBSDE model is calculated as:

$$\rho_1 = \underbrace{1+d}_{u(0,\xi), \nabla u(0,\xi)} + \underbrace{(N-1)(2(d+10) + 2(d+10) + 2d)}_{\text{batch normalization layers}}$$
$$+ \underbrace{(N-1)(d(d+10) + (d+10)^2 + d(d+10))}_{\text{fully connected layers}} \tag{4}$$

Comparing (3) and (4), one immediately notice that:

1. DBSDE uses one parameter to approximate $u(0,\xi)$ and $d$ parameters for $\nabla(0,\xi)$. We do not have those two sets of parameters.

2. The network proposed by DBSDE is a MLP stacked $N$ times where $N$ is the time steps that discretize the temporal dimension, which leads to $\rho_1 \sim \mathcal{O}(Nd^2)$ complexity, whereas $\rho_0 \sim \mathcal{O}(d^2)$ in our model. Recall that $N$ is a hyperparameter that needs to be tuned case by case. Therefore, having the network complexity controlled by $N$ poses numerical challenges when $N$ is large. Our model has no such constraint.

 ## 3.2 Faster Convergence Rate

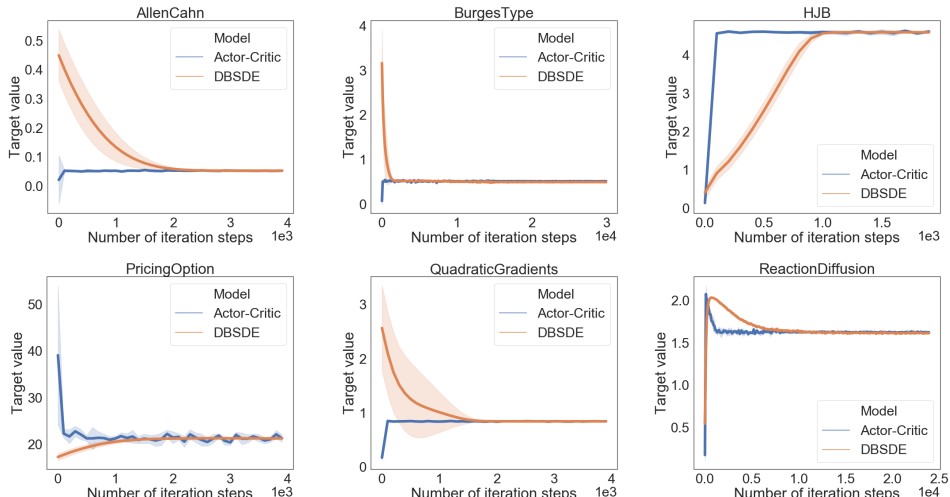

Figure 2: Evolution of the target solution $u(0, \xi)$ during training.

Compared to DBSDE, it is noteworthy that our model needs much fewer iterations to converge in general. Intuitively, this could be attributed to the fact that our neural network is shallower than DBSDE by N times, which naturally requires fewer samples to train. Recall the fact that our run-time per iteration is also shorter, together they can explain why our algorithm is significantly faster than DBSDE as previously discussed around Table 1. In addition to the convergence rate speedup, one also notices a significant drop in the variance. The important contributing factor is the actor critic architecture which by nature is an effective variance reduction technique.

## 3.3 Fewer tuning hyperparameters

DBSDE uses one trainable parameter to fit the solution $u(0, \xi)$, which assumes a probability distribution in a predefined region $(x_a, x_b)$. Thus $x_a$ and $x_b$ are two hyperparemters that need to be chosen case by case. In comparison, we use the critic network to parametrize $u(0, \xi)$. It is arguable that the network design is a hyperparameter by itself, but in practice, we use the same critic network structure with the same initialization procedure (xavier-uniform) in solving all the equations in table 1 and all achieved higher accuracy level than DBSDE. To some extent, the experiments suggest that the critic network, with initialization process properly designed, applies regularizing to $u(0, \xi)$ automatically.

## 4 Discussion and conclusion

The limitation of our model, and in fact of all existing deep learning models for high dimensional PDEs, is only focusing on learning mappings between finite-dimensional spaces. Therefore, one needs to perform training every time the solution is to be evaluated at a new point. In practice this can be computationally expensive as the solutions are typically desired at a large collection of points. A future direction to lift the limitation is to generalize the neural networks proposed in this work to "neural operators" that learn mappings between function spaces. In 2-d and 3-d scenarios, the pioneering works of Li et al. (2021) and Liu et al. (2021) show that neural operators allow accurate transfer learning and even zero-shot super-resolution. However, due to the curse of dimensionality, simply generalizing their approach to high dimension scenarios is not feasible.

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
