# OpenReview forum: "Actor-Critic Algorithm for High-dimensional PDEs"
_NeurIPS.cc/2021/Workshop/DLDE — DLDE Workshop -- NeurIPS 2021 Poster_

### Official Review · Reviewer_QEo4 · 2021-10-11

**Confidence:** 3

**Review:**

- Summary

This method proposes a new approach to solving high-dimensional PDEs using neural networks. The method is time-independent and as such does not scale in computational complexity with the number of time-steps used. It is also claimed that it has better convergence properties and fewer hyperparameters to tune.

- Main review

The authors introduce the abbreviations D(BSDE) without ever defining them of explicitly explaining what they refer too. This and other issues of clarity in the introduction should definitely be addressed to make sure the the text is accessible to a wide audience. Noticeably, the introduction has no reference to previous work, and the first time any relevant previous work is referenced is on page 3.

The discussion of the method itself in Section 2 could be made clearer (there is still space in the submission so it shouldnt be an issue to expand this a bit). A way to make this clearer would be to relate the general definition to a concrete example (e.g., possibly from the PDEs studied in the experimental section). Moreover, once again the reader is expected to be familiar with details of DBSDE, which has not been explicitly introduced or explained.

Moreover, even though it is central to the paper, I do not believe the analogy to actor-critic architectures is properly explained. Since this is an idea that is referred to many times it would be important to explain and make the connection clearer.

If the authors feel space is insufficient for these updates, an appendix can always be used.

It is nice that the authors use a similar evaluation to previous work, with the same equations and computational environments.

Despite the issues above, the paper seems to propose an interesting idea, with some experiments to demonstrate its promise. I believe it is at an acceptable level for acceptance to the workshop, especially if the authors address some of the issues mentioned above.

**Score:**

3: Good paper

---

### Official Review · Reviewer_BSTy · 2021-10-12
**Interesting idea, but unclear research gap**

**Confidence:** 4

**Review:**

Summary
The paper proposes a neural network-based model to learn solutions to parabolic backward stochastic differential equations. The proposed work claims to decrease the number of trainable parameters, hyperparameters, and convergence rate. While the general approach seems promising, the paper does not seem to articulate the addressed research gap and the benefit of the proposed method remained unclear in the broader context of learning-based PDE solvers.

Review:
Pro:
- The paper clearly states the intended contributions and targets the interesting question of how deep RL algorithms can be used to learn quickly-converging PDE solvers.
- The mathematical problem statement is quite clear.

Improvements:
- First and foremost, the paper significantly lacks references. The lack of references introduces an unclarity whether the paper is targeting an important research gap and contributing towards the field of learning-based PDE solvers. The authors could greatly improve the work by adding references to statements such as "largely reduced trainable paremeters from O(Nd^2) to O(d^2)" or using a single network instead of a "sequence of [...] multilayer feedforward networks" [56-57].
- Second the approach remained relatively unclear to me. It seems to me that the authors propose to solve nonlinear parabolic PDEs by reformulating them into a discretized Markovian SDE. The solution to the SDE is then learned by two shallow time-independent fully-connected networks that predict the solution at t=0 and t, respectively. It remained unclear to me 1) whether the Feynman-Kac is applicable to all parabolic PDEs in the form of Eq. 1 and why Feynman-Kac is the best way to re-formulate the given PDE into an ODE (which could be solved by adding a reference), 2) why we need different networks for t=0 and t=t, 3) pros/cons of pulling the time out of the neural network input (e.g., error might accumulate). Further, the parallel of the proposed approach an actor-critic method could be explained (60-61). And, it would help to reference Fig 1 earlier in the text to guide the reader along the systems architecture diagram.
- Third the paper seems to overstate contributions. Not only are the listed contributions mostly unreferenced, but also the title claims applicability to general "high-dimensional PDEs", whereas the work only describes parabolic backward differential equations. The results section only shows comparative results with DBSE and could be improved by 1) clearly stating the significance of the differences between DBSE and the proposed method, 2) providing an argument why DBSE has been chosen as only baseline, and/or 3) adding other baselines such as common PINNs.


Minor:
- It would be beneficial if the authors clearly define "high-dimensional". I assume they are referring to the number of spatial dimensions (not the number of variables, spatial, or temporal discretization points).
- The results section could be improved by plotting errors.
- A quick google shows that this work might be interesting to read and reference: Pan et al., Reinforcement Learning with Function-Valued Action Spaces for Partial Differential Equation Control, PMLR, 2018
- The intro sentence (line 16-17) state that the proposed approach "address[es] those issues", but it is unclear which issues the approach addresses. This could be improved by simply writing "adresses to solve parabolic PDEs with many spatial variables that can be reformulated into a BSDE".

Thank you for the read!

**Score:**

1: Reject: trivial or wrong

---

### Official Review · Reviewer_ntPj · 2021-10-12

**Confidence:** 2

**Review:**

This paper proposes to solve high-dimensional nonlinear parabolic partial differential equations by using an Actor-Critic neural network architecture. The performance benefits are well articulated although a bit repetitive.

The paper lacks background and literature review, with a corresponding lack of references.
The acronyms DBSDE should be defined. Otherwise it is difficult to understand the comparisons.
Fig 1 should be described in text.
In Fig 2, in all the legends, the “Model’ can be mistaken for another model. The plots should be rescaled since most of the plot area is after convergence.
For numerical results, the proposed method should also be compared to the solution and/or non deep learning approaches to solving the PDE systems.

The potential for the proposed idea is difficult to realize because the paper lacks essential details. While some components are well presented, other key components expected from a research paper are missing.

**Score:**

3: Good paper

---

### Decision · Program_Chairs · 2021-10-16

**Decision:**

Accept (Poster)

**Comment:**

Reviewers were mixed about this paper. The stand-out concern is clearly the fact that this paper does very little to set itself in context. I would echo these concerns, and this is clearly the single greatest weakness of the paper. For example I would highlight lines 54--62, in which the proposed technique is compared only against some nebulously-defined "other".

Despite these concerns, I am inclined to agree with reviewer QEo4 that the paper is acceptable for a workshop, and would strongly encourage the authors to address the concerns raised by reviewers.

In addition to the review comments, a few comments of my own:

- Equation (2) assumes that the SDE is solved via the Euler--Maruyama method. Whilst analytically convenient, I believe it is more elegant to treat the general case for as long as possible (without reducing to just the EM method at this early stage). If the noise has particular structure then it may become desirable to solve the SDE via some more efficient numerical method, e.g. Milstein if using commutative noise or Heun if using additive noise (Heun converges to the Stratonovich solution, but for additive noise then Itô and Stratonovich are identical).
- I think equation (2) may be missing a $t_{n+1}-t_n$ coefficient for the $f$ term.
- The "not too far" of line 67 is usually referred to as a "trust region".